# VisIT-Bench: A Benchmark for Vision-Language Instruction Following Inspired by Real-World Use

**Yonatan Bitton**[*,1,2]  **Hritik Bansal**[*,3]  **Jack Hessel**[*,4]  **Rulin Shao**[5]  **Wanrong Zhu**[6]
**Anas Awadalla**[5]  **Josh Gardner**[5]  **Rohan Taori**[7]  **Ludwig Schimdt**[4,5,8]

## Abstract

We introduce VisIT-Bench (**Vis**ual **Ins**Truction **Bench**mark), a benchmark for evaluating instruction-following vision-language models for real-world use. Our starting point is curating 70 "instruction families" that we envision instruction tuned vision-language models *should* be able to address. Extending beyond evaluations like VQAv2 and COCO, tasks range from basic recognition to game playing and creative generation. Following curation, our dataset comprises 592 test queries, each with a human-authored instruction-conditioned caption. These descriptions surface instruction-specific factors, e.g., for an instruction asking about the accessibility of a storefront for wheelchair users, the instruction-conditioned caption describes ramps/potential obstacles. These descriptions enable 1) collecting human-verified reference outputs for each instance; and 2) automatic evaluation of candidate multimodal generations using a text-only LLM, aligning with human judgment. We quantify quality gaps between models and references using both human and automatic evaluations; e.g., the top-performing instruction-following model wins against the GPT-4 reference in just 27% of the comparison. VisIT-Bench is dynamic to participate, practitioners simply submit their model's response on the project website; Data, code and leaderboard is available at https://visit-bench.github.io/.

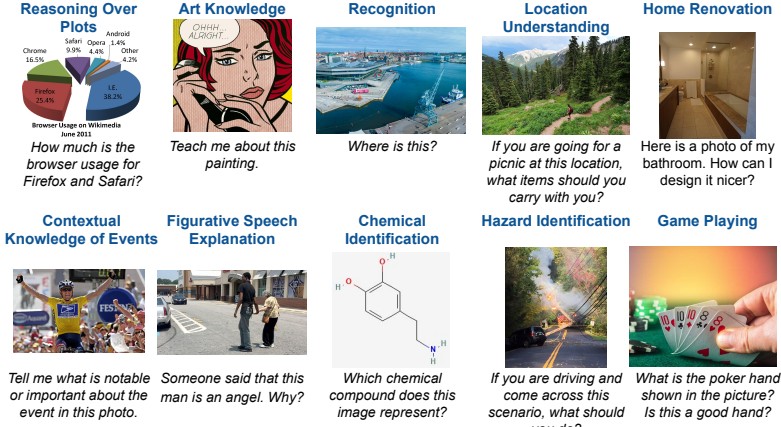

Figure 1: A sample from the 70 instruction families in VisIT-Bench representing tasks we envision instruction-following vision-language models *should* be able to follow.

[*]Equal contribution. [1]Hebrew University  [2]Google Research  [3]UCLA  [4]Allen Institute for AI  [5]University of Washington  [6]UCSB  [7]Stanford  [8]LAION ,

37th Conference on Neural Information Processing Systems (NeurIPS 2023) Track on Datasets and Benchmarks.

# 1 Introduction

A long-standing challenge for artificial intelligence is to build general-purpose assistants that can, in collaboration with humans, solve diverse and never-before-seen tasks [1]. For textual tasks, several recent works [2, 3, 4, 5, 6, 7] have shown that fine-tuning language models such as GPT-3 and LLaMA with supervised instruction+response examples [8, 9, 10] enables them to respond to imperative requests and questions without task-specific training. Zero-shot generalization is promising not only for standard academic benchmarks, but – perhaps more-so – for creative, useful, and real-world queries that downstream users of language technologies are likely to make.

On the multimodal side, recent instruction-following vision-language models also provide a zero-shot interface. Given an image (or multiple images) and a query (e.g., "how many apples are in this image?" or "What is this?" or "Write a poem in the style of Robert Frost about this scene.") a textual response is provided. Recent works like OpenFlamingo [11, 12], LLaVA [13] and others [14, 15, 16, 17, 18], have implemented this interface with promising initial results. Although standard benchmarks like VQAv2 [19] and COCO captioning [20] are commonly used to assess performance, less is known about how models perform on broader, open-ended queries that resemble real-world user behavior. Evaluations of such queries typically rely on informal and qualitative approaches.

To support quantitative evaluation for this setting, we present VisIT-Bench (**Vis**ual **Ins**Truction **Bench**mark), a dynamic benchmark consisting of 592 challenging vision-language instructions. Each instance contains an instruction, input image(s), a instruction-conditioned caption (a human-crafted caption for the image(s)/instruction), and a human verified reference (Figure 2). Instructions are image-contextual imperative requests or questions, e.g., for an image of pancakes, a user asks *"how can I cook this in a healthy way?"*. Different from existing zero-shot evaluations, many of the instructions focus on open-ended generation requests (e.g., *"write a poem..."* or *"what should I bring if I were to visit here?"*).

We created VisIT-Bench to cover a wide array of "instruction families". Our starting point was a set of 70 "wish-list" tasks such as "home renovation" and "gardening tips" collected by the authors: each requiring varied high-level skills from recognition to complex reasoning (Figure 1). We derived 25/70 instruction families from benchmark tasks such as Visual Question Answering (VQA) [21] and robust change captioning [22] into a chatbot-style format (this reformatting differs from prior work [14, 17, 13], as we focus on open-ended chatbot style responses.). Notably, 10 of these repurposed tasks involve multiple images.

We started with 10 images for each instruction family. Our annotators, guided by an example, create a new instruction, and provide a (permissively licensed) image. For each instruction, we next collect instruction-conditioned captions – unlike prior work [23, 24] these descriptions are designed not only to describe the image in general, but also, surface information targeted to the instruction. Finally, we use instruction-conditioned captions to generate a reference candidate output from GPT-4; an additional human verification step discards GPT-4 references deemed to be incorrect.

We conduct a large-scale empirical comparison of multimodal instruction-following models using VisIT-Bench (§4). We first gather predictions for each instance from 7 candidate models. Then, we collect 5K human judgements of output quality by pitting model outputs head-to-head, and (in a forced-choice setup) crowd-sourcing pairwise preference judgements. This analysis not only reveals

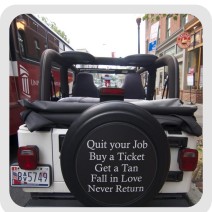

| | **Instruction** | **Instruction-Conditioned Caption** | **Human-Verified GPT4 Response** |
|---|---|---|---|
| | Create a catchy title for a country song based upon the advice printed on the wheel cover. | There is a jeep parked on the street with the top down. There is a wheel cover on the spare tire in the back with an inscription that says, "Quit your Job, Buy a Ticket, Get a Tan, Fall in Love, Never Return." | "Tan Lines & Ticket Stubs: A Love Story Unbound" - A country song about leaving it all behind, finding love, and embracing the freedom of the open road. |

Figure 2: An example from VisIT-Bench displays an image, instruction, an instruction-conditioned caption based on the instruction, a GPT-4 suggested response, and a label confirming its accuracy. All 678 entries in VisIT-Bench have such labels, with 592 confirming accurate GPT-4 responses. These components aid in assessing multimodal chatbots and updating a dynamic leaderboard.

Table 1: Comparison with related works for evaluating instruction-following vision-language models. Win-rates* refers to the model win-rates against a reference output/model.

| | MultiInstruct [32] | Owl [17] | InstructBLIP [14] | M³IT [33] | LVLM [34] | GAVIE [35] | **VisIT-Bench** |
|---|---|---|---|---|---|---|---|
| Number of Models | 1 | 5 | 3 | 4 | 8 | 5 | 10 |
| Number of Skills Tested | 9 | 6 | 13 | 13 | 47 | 16 | 70 |
| Multiple-Images | ✗ | ✓ | ✗ | ✗ | ✗ | ✗ | ✓ |
| Video | ✗ | ✗ | ✓ | ✓ | ✗ | ✗ | ✗ |
| Multi-Turn Conversations | ✓ | ✓ | ✓ | ✓ | ✓ | ✗ | ✗ |
| Multilingual Conversations | ✗ | ✓ | ✗ | ✓ | ✗ | ✗ | ✗ |
| Instruction-conditioned Captions | ✗ | ✗ | ✗ | ✗ | ✗ | ✗ | ✓ |
| Chatbot-style Responses | ✗ | ✗ | ✗ | ✗ | ✗ | ✗ | ✓ |
| Dataset-specific Evaluation | ✓ | ✓ | ✓ | ✓ | ✓ | ✗ | ✗ |
| Human Evaluation | ✗ | ✓ | ✗ | ✗ | ✓ | ✗ | ✓ |
| Auto/GPT-4 Evaluation | ✗ | ✓ | ✗ | ✓ | ✗ | ✓ | ✓ |
| Win-rates* | ✗ | ✓ | ✗ | ✓ | ✗ | ✓ | ✓ |
| Elo Rating | ✗ | ✗ | ✗ | ✗ | ✗ | ✗ | ✓ |

significant differences between models (e.g., that LLaVA-13b [13] is generally preferred to Panda [18]), but also, that the human verified references in our corpus are preferred significantly more than the ones generated using multimodal models. We summarize head-to-head comparisons with two metrics: 1) Elo ratings [25, 26], which provide *relative* "skill" rating estimates encoding the probability that model A will be preferred to model B; and 2) win rate versus our references, which provides an *absolute* metric. The best model according to human judgement is LLaMA-Adapter-v2 [16], yet it only wins in a pairwise setting against the reference in 27.4% of cases.

Finally, we design an automated evaluation for VisIT-Bench, utilizing GPT-4 to rank pairs of model responses based on factors like correctness, relevance, and fluency. Using the instruction-conditioned caption and the instruction, GPT-4 determines the better response between two options, expediting iteration compared to human preferences. We explore *reference-free* and *reference-backed* versions of this metric. Compared to various metrics (BLEU-4 [27], ROUGE-L [28], METEOR [29], CIDEr [30], and BERTScore [31]), our evaluation aligns best with human preferences. For example, it achieves a 94% agreement rate in the cases where all five annotators agree. Figure 6 illustrates the process.

While it is difficult to *a priori* envision all possible scenarios under which more performant multimodal chatbots might be used, we hope VisIT-Bench can provide a path to improving vision-language models "in the wild." Table 1 presents a summary of our contributions in comparison to the recent works [32, 14, 17, 33, 34, 35] in the evaluation of multimodal chatbots. We publicly release VisIT-Bench data, code, and automatic metrics in https://visit-bench.github.io/.

## 2 VisIT-Bench: A Real-World Inspired VL Instruction-Following Benchmark

VisIT-Bench was built to emulate real-world applications of multimodal models through image-text tasks, creating an extensive and practical benchmark. These tasks, or 'instruction families', are seen as key capabilities of a high-performing vision-and-language model. Although our selections are not exhaustive, they provide a broad basis for evaluating beyond academic benchmarks. We prioritize family coverage vs. number of instances-per-task. The final corpus, comprising 678 instances and 1,159 public images, can be found at VisIT-Bench Sheet Multi-Images. VisIT-Bench instances are either from 45 newly assembled instruction families or reformatted from 25 existing datasets (see Table 5). Notably, 10 instruction families cater to *multi-image* query scenarios (e.g., Figure 4).

### 2.1 Data Collection

The authors of this work perform an initial annotation step of curating instruction families. For each instruction family not derived from an existing task (45 out of 70), we designate a name for the family (e.g., "Contextual Knowledge of Events") and identify an image-instruction pair that exemplifies the category, along with a sample response ("Martin Luther King Jr. is waving to acknowledge and greet the crowd of protesters [...]"). 10 sample famiies are in Figure 1.

We work with crowdworkers at $18/hour to execute the annotation steps, as outlined in Figure 3: (1) taking the image/instruction example as a guiding seed task crowdworkers formulate a new instruction that examines the same instruction family ("instruction generation"); (2) crowdworkers create detailed image captions that describe the image and allow an entity, relying solely on this

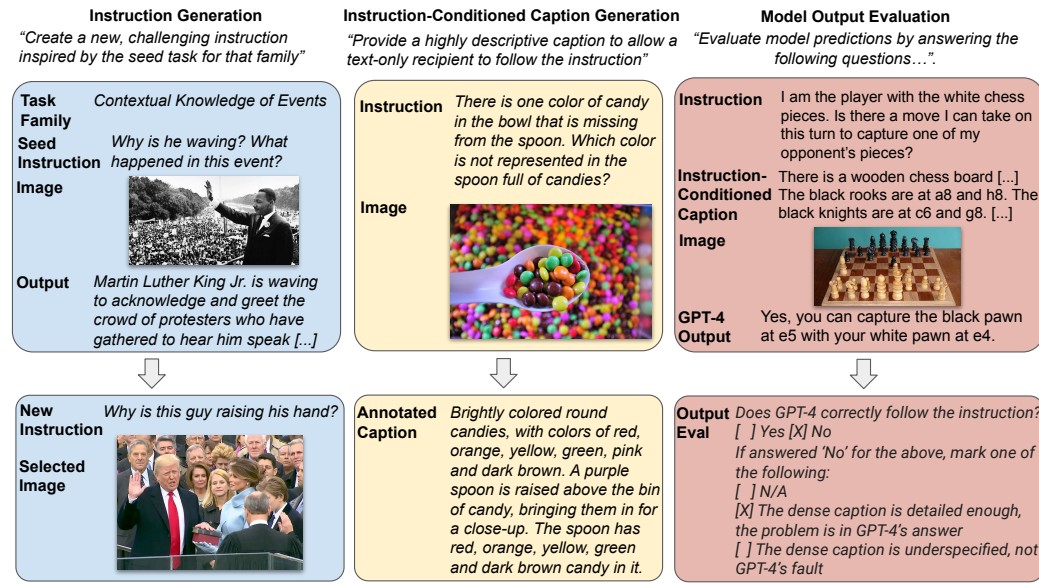

Figure 3: Data collection steps: (1) **Instruction Generation** from a seed task (left). (2) **Caption Generation** creates rich *instruction-conditioned captions* for GPT-4 (middle). (3) **Model Evaluation** with human-validated GPT-4 responses (right). Top: rater instructions; bottom: outputs.

text, to interpret and execute the given instruction successfully ("instruction-conditioned caption generation"); (3) crowdworkers assess the correctness of GPT-4's response to the instruction ("model output evaluation"). We further elaborate on these steps using human annotators below.

**Re-formatting existing datasets.** 25/70 instruction families (corresponding to 25*10=250 instances) are re-formatted versions of existing vision-language tasks (See Appendix D for full list).This process involves re-formatting tasks into chatbot-style instruction/response versions. In re-formatting, we re-write instructions to retain the original task's goal while maintaining the original images, see Figure 4. These repurposed tasks are integrated into our data collection process, ensuring uniformity between the chatbot-style answers in the full VisIT-Bench instances and the reinterpreted tasks.

**Instruction Generation.** Here, annotators create a new instance from the same instruction family as a given example, along with an instruction and corresponding image. For instance, in Figure 3 (left), the instruction family is "Contextual Knowledge of Events", and the example instruction is *"Why is he waving? What happened in this event?"* alongside an image of Martin Luther King, Jr. To collect images, annotators were instructed to use openverse for Creative Commons licened images.

**Instruction-Conditioned Caption Generation.** Annotators are provided with the image and instruction, and are tasked to construct a caption that is rich enough to allow an entity, solely receiving the text they author, to follow the instruction. These captions, termed *instruction-conditioned captions*, aid GPT-4 reference generation and text-only evaluation. See Figure 3 (middle) for an example: an annotator doesn't just mention the skittles and a spoon, but, given the query regarding specific colors, they indicate the exact colors in detail.

**Model Output Evaluation.** The goal of this stage is to gather human-validated reference chatbot responses for each multimodal instruction query. We initially obtain response candidates from GPT-4 given the instruction and the instruction-conditioned caption. GPT4's prompt is: *"Consider an image depicted by: <caption>'. Now, briefly follow this instruction, and you can add a short explanation: <instruction>'. Response:* This prompt is employed for both single and multiple image instances, with appropriate modifications for the latter. Then we verify each response with human annotators. If a response is marked incorrect, the annotator identifies whether the issue lies with the detail level of the instruction-conditioned captions or with GPT-4's response itself. For VisIT-Bench, we discard any case marked as incorrect for either reason. An example is given in Figure 3 (right), where GPT-4's

**NLVR2**

The In this task you will be provided with two individual images i.e., BEFORE and AFTER. Please study them carefully and determine if the following sentence is correct (Answer "Yes"/"No"):

None of the dogs in these images are wearing clothing.

**1** 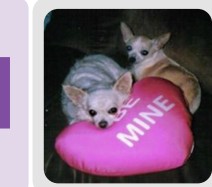 Two small chihuahuas, one with all white fur, and the other with a light brown coat of fur on its back, but white fur on its head and neck, are near a small pink pillow that has large white words proclaiming, "BE MINE" on it. The all white chihuahua is resting its head on the pillow, while the other one is laying behind it.

**2** 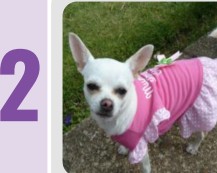 A small all white chihuahua is standing and looking straight at the camera. It's wearing a pink and white polka-dotted dress, over which is a pink shirt with some white text on it. Under the dog is a cement walkway, and there is grass to the dog's right and to the right of the walkway.

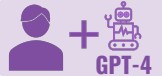 No. The second image has a dog wearing a pink and white polka-dotted dress and a pink shirt with white text on it.

Figure 4: An example multi-image task from VisIT-Bench, sourced from NLVR2 [36], tests visual reasoning. While NLVR2 uses a sentence, two images, and a binary answer, we add a zero-shot prompt, a instruction-conditioned caption per image, and a verified GPT-4 reply. This chatbot-style design aids automatic evaluation of future chatbot interactions.

candidate reference response aims to answer a question about a chess position (which it does so incorrectly, and thus, the instance is discarded).

## 2.2 Data Collection Annotation and Results

We conduct the data collection steps in Figure 3 using Amazon's Mechanical Turk (MTurk) platform. Prior to annotating, each MTurk worker passed a qualification test, which involved five to ten sample tasks designed to assess their ability to generate high-quality annotations. More detailed information about the execution process and full user interface examples can be found in Appendix C.

Annotation results are in Table 2. We assess our collection and filtration efficiency. For single-image tasks, our pipeline's yield was 91.5% from the original candidate set. However, the success rate dropped to 63.0% for multi-image tasks, accompanied by an uptick in issues either in the captions (6.0%) or GPT-4's responses (30.0%). This drop suggests that multi-image queries may pose a more difficult data collection challenge.

Table 2: Human rating metrics for the VisIT-Bench dataset: overall, single-, and multi-image tasks.

| Metrics | Overall | Single | Multi |
| --- | --- | --- | --- |
| GPT-4 Correct (%) | 87.3 | 91.5 | 63.0 |
| Problem in Caption (%) | 4.0 | 3.6 | 6.0 |
| Problem in GPT-4 (%) | 7.7 | 3.8 | 30.0 |

## 3 VisIT-Bench Analysis

We analyze the tasks, images, and instruction-conditioned captions of VisIT-Bench.

**Are instruction-conditioned captions necessary?** To elucidate instruction-conditioned captions' role, we conduct an experiment on 150 single-image instances. We replace our instruction-conditioned captions with BLIP2 [15] captions, a leading image captioning model, and feed them to GPT-4 for a chatbot response. See Figure 5 for this process. We manually evaluate whether the resulting output accurately followed the instructions. We find instruction-conditioned captions yielded 91% correct results, but with BLIP2 captions, success dropped to 31% (Table 2). This underscores the importance of instruction-conditioned captions in the construction of VisIT-Bench, and shows that the instances in our dataset are sophisticated enough such that most are not solvable by using a simple Socratic model [37] baseline of caption → LLM.

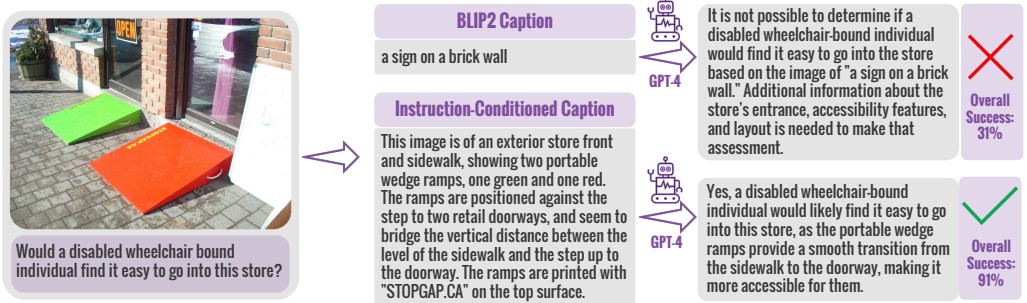

Figure 5: This experiment evaluates the value of instruction-conditioned captions in accurate instruction-following tasks. Given an image and instruction, GPT-4 generates responses using both a instruction-conditioned caption and a less detailed BLIP-2 [15] caption. The latter's imprecision leads to an error, emphasizing the need for detailed, task-specific captions.

**What skills are required for VisIT-Bench?** The full list of instruction families we cover are in Appendix Table 6. Following [38], for the VisIT-Bench instructions, we extract the most frequent root verbs and their direct nouns (a full plot is in Figure 10). The most common include: *'answer question'*, *'write story/poem'*, *'create title'*, etc. There's also a long-tail of diverse requests that demand comprehension, commonsense, and cross-modal understanding, e.g., *'identifying objects'* to *'need ingredient'* to *'connect device'*. Additional examination reveals a range of underlying skills required ranging from *'emotion identification'* to complex reasoning tasks such as *'paper folding'*.

**What is contained in VisIT-Bench images?** We detect all the COCO [20] objects present in the images from our dataset using Yolov5-L [39]; The most common detected objects in VisIT-Bench are "person" (~ 900 detections), chair, and car (~ 100). But, a long tail of rarer objects exists as well: full distribution in Appendix Figure 9. Overall, to perform well at VisIT-Bench, a model must account for a broad range of scenes and objects.

## 4 Experiments

We evaluate a range of state-of-the-art publicly accessible vision-and-language chatbots on the 592 instances in VisIT-Bench. In §4.1, we provide the details of the instruction-following models in our benchmark. Following this, we collect the human preferences for pairwise model generations to achieve a human-guided Elo ranking and the win-rates against the reference of the models in §4.2. We then develop automatic evaluation on VisIT-Bench in §4.3, that can be scaled and improved given new and improved models. Finally, we establish the trustworthiness of our automatic evaluation method by performing agreement analysis with the human judgments in §4.3

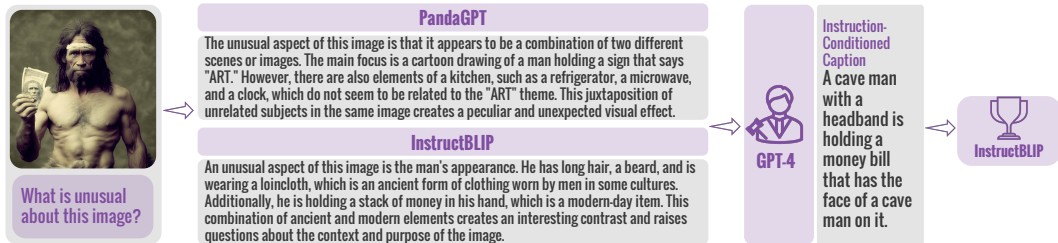

Figure 6: ELO-based evaluation for VisIT-Bench: Our reference-free approach uses a GPT4 evaluator to compare two instruction-following models with an instruction and a instruction-conditioned caption. The instance is obtained from an existing dataset, WHOOPS! [40].

Table 3: Human scores for models are displayed as ELO ratings and win-rates against the reference, summarizing 5.0K pairwise judgments. The 'matches' column shows each model's participation count, and 'win-rate vs. reference' denotes its win rate against reference outputs.

| | Model | Elo | matches | Win-rate vs. reference (w/ # ratings) |
|---|---|---|---|---|
| Single Image | Human Verified GPT-4 Reference | 1223 | 1439 | – |
| | LLaVA (13B) | 1085 | 1462 | 26.23% (n=244) |
| | LlamaAdapter-v2 (7B) | 1061 | 1507 | **27.41%** (n=259) |
| | mPLUG-Owl (7B) | 995 | 1345 | 14.95% (n=214) |
| | InstructBLIP (13B) | 957 | 1315 | 12.37% (n=194) |
| | MiniGPT-4 (7B) | 893 | 1513 | 14.72% (n=299) |
| | PandaGPT (13B) | 786 | 1441 | 10.48% (n=229) |
| Multiple Images | Human Verified GPT-4 Reference | 1193 | 210 | – |
| | mPLUG-Owl | 997 | 190 | 15.38% (n=78) |
| | Otter v1 | 917 | 147 | 3.17% (n=63) |
| | OpenFlamingo v1 | 893 | 171 | 4.35% (n=69) |

## 4.1 Models

We evaluate LLaVA-13B [13], InstructBLIP-13B [14], MiniGPT4-7B [41], mPLUG-Owl-7B [17], LlamaAdapter-v2-7B [16], PandaGPT-13B [18], VisualChatGPT [42], Multimodal GPT [43], Open-Flamingo v1 [11, 44] and Otter v1 [45]. For the execution-based VisualChatGPT [42], we implement a chat window for each sample, hold inputs and intermediate chains of thoughts and actions in memory, and feed the images and the instruction sequentially. For OpenFlamingo [11] and Otter [45], we feed the image(s) and the instruction in an interleaved format. For the others, we feed the image to the vision feature extractor and feed the instruction as a prompt to the text encoder.

## 4.2 Human Evaluation

We collect 5K pairwise human preference judgements across an initial set of 6 models and the human-verified references. For 1K uniformly randomly sampled tuples of (query, model A, model B), we collect 5 crowdworker judgements each. Preferences are collected in a "forced choice" setting, annotators are instructed to decide based on accuracy, helpfulness, and detail. We provide the template for the human annotation process in Appendix Figure 15. We summarize the results with two metrics:

**Relative metric: Elo** We follow [26] and compute Elo ratings, treating each pairwise human judgement as a "match." The difference between the Elo ratings of two different models provides an estimate for the win probability when pitting model A vs. model B. More details are in Appendix E.

**Absolute metric: Win rate vs. reference.** We provide a win-rate vs. the human-verified reference. We use the 1.4K pairwise human judgments where one of A or B is the reference. We report the percent of cases where the human judge prefers the output from that model vs. the human-verified GPT-4 reference output. Because we do not allow for ties in our forced-choice setup, if the annotator believes the responses are of equal quaity, they choose one arbitrarily.

**Results** Table 3 contains the Elo and win-rate vs. reference. In terms of Elo, the Human Verified GPT-4 reference achieves a higher rating than all alternatives, validating the quality of our reference set: concretely, for our Elo settings, the reference (Elo =1223) has an estimated win-rate over one of the best performing models, LLaVA, (Elo =1085) of 69%, and an estimated win rate of 93% against the lowest performing model in this setup, PandaGPT (Elo =786). This result can partly be explained by the training process of the underlying models: The improved performance of LLaVA (13B) might be attributed to its fine-tuning process, which utilized 150K instruction-tuning data that is rich in both diversity and quality. Interestingly, despite achieving a slightly lower Elo (the computation of which is based on *all* head-to-head "matches", rather than just ones against the human reference), LlamaAdapter-v2 (7B) wins with the highest rate against the reference. However, the complexity of models and tasks in VisIT-Bench makes it challenging to definitively pinpoint the factors influencing performance. We conduct an initial exploration of this result in Section 4.3.

Table 4: As of July 19th, 2023, reference-free Elo rankings summarize 12K matches between models, each with 2 GPT-4 queries. With the dynamic VisIT-Bench, rankings update as more models join the leaderboard and more head-to-head evaluations occur.

|  | Model | Elo | matches | Win vs. Reference (w/ # ratings) |
|---|---|---|---|---|
| Single Image | Human Verified GPT-4 Reference | 1370 | 5442 | - |
|  | LLaVA (13B) | 1106 | 5446 | **17.81%** (n=494) |
|  | LlamaAdapter-v2 (7B) | 1082 | 5445 | 13.75% (n=502) |
|  | mPLUG-Owl (7B) | 1081 | 5452 | 15.29% (n=497) |
|  | InstructBLIP (13B) | 1011 | 5444 | 13.73% (n=517) |
|  | Otter v1 (9B) | 991 | 5450 | 6.84% (n=512) |
|  | VisualGPT (Da Vinci 003) | 972 | 5445 | 1.52% (n=527) |
|  | MiniGPT-4 (7B) | 921 | 5442 | 3.26% (n=522) |
|  | OpenFlamingo v1 (9B) | 877 | 5449 | 2.86% (n=524) |
|  | PandaGPT (13B) | 826 | 5441 | 2.63% (n=533) |
|  | Multimodal GPT | 763 | 5450 | 0.18% (n=544) |
| Multiple Images | Human Verified GPT-4 Reference | 1192 | 180 | - |
|  | mPLUG-Owl | 995 | 180 | 6.67% (n=60) |
|  | Otter v1 | 911 | 180 | 1.69% (n=59) |
|  | OpenFlamingo v1 | 902 | 180 | 1.67% (n=60) |

## 4.3 Automatic Evaluation and Leaderboard

Because it is costly to gather human pairwise preference judgements for new model submissions, to support faster model development, we seek an automatic evaluation procedure that produces high correlation with our human evaluation setup.

**Automatic evaluation metric candidates.** We consider several existing reference-backed evaluation metrics: BLEU-4 [27] , ROUGE-L [28], METEOR [29], CIDEr [30], and BERTScore [31], we use the RoBERTa-Large english version [46], treating the human-verified GPT-4 reference as the evaluation reference. We additionally report two baseline metrics: random, which assigns a random score without accounting for the candidate, and length, which assigns a score equal to the number of non-whitespace tokens in the candidate. Beyond existing metrics and baselines, following the recent line of work utilizing API-accessed LLMs with a prompt for automatic evaluation [6, 47], we consider two GPT-4[1] [7] backed evaluation metrics.

Specifically, we provide the LLM with: 1) a system prompt describing the desired evaluation behavior; 2) the instruction-conditioned caption for the image; 3) the instruction to be followed;

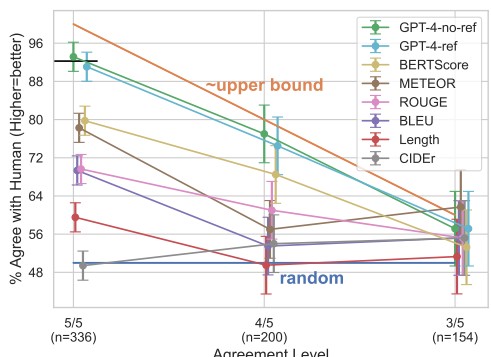

Figure 7: Correlations between evaluation metrics and human preferences ranked by performance, with our reference free evaluation (GPT-4-no-ref) showing the strongest alignment. Bottom: random chance (50%), top: upper performance bound.

and 4) two candidate generations dubbed "Response A" and "Response B". We also consider a reference-backed version where the human-verified reference is provided as well. We provide our prompts in Appendix F. To mitigate potential biases in "A" and "B" positioning, for all pairs of candidates, we run two queries covering both possible orderings. Our prompt encourages the model to think step-by-step so that its chain-of-thought process is made explicit [48, 49]. Despite strongly encouraging the model to select between the two references in a forced-choice setup, it sometimes

---
[1]OpenAI [7] hosts several API versions of GPT-4 and updates them over time, we use the versions they host interchangeably (specifically, our evaluations mix their models named: `gpt-4-0314` (which became depreciated during the development of this work) and `gpt-4` (which underwent an update during our experiments).

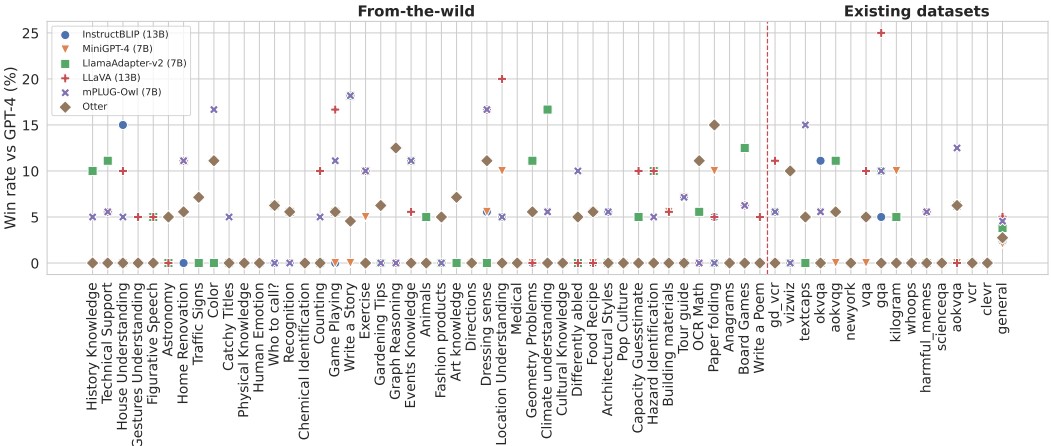

Figure 8: Reference-free assesment win rate vs. human-verified GPT4 response for each instruction category. Axes: win rate (Y), instruction categories (X). Categories are from-the-wild or existing datasets. VisIT-Bench facilitates analysis of diverse instruction tuning tasks.

refuses and outputs "tie" which we account for later. We call the reference-free version of this metric "GPT4-no-ref", and the reference-backed version of this metric "GPT4-ref".

**Evaluating evaluation metrics.** We measure the correlation between the candidate metrics and human judgements using a pairwise framework. Specifically, we use a subset of the 5K pairwise human judgements in § 4.2. For 690 pairwise instances where both candidate instances are model-generated (rather than human-verified references), we have 5 pairwise judgements from crowd-workers. For 336 pairs, there is 5/5 agreement, for 200 pairs, there is 4/5 agreement, and for 154 pairs, there is 3/5 agreement. For each metric, we measure the percent of time the metric is able to accurately reconstruct a majority vote judgement from the 5 crowdworkers. The newly proposed GPT-4 based metrics sometimes outputs "tie" (this happens in 10-15% of cases overall) – for fair comparison with the other metrics in forced choice setting, we randomly choose one of the two options when GPT-4 reports a tie.

The results are in Figure 7, with GPT-4-no-ref best aligns with human correlation. The best performing metric is our newly proposed GPT-4 based metric, which accurately reconstructs majority-vote pairwise human judgments better than alternatives ($p < .05$; binomial proportion CI nonoverlapping). For example, for instances where 5/5 annotators agree, GPT4-no-ref, with no reference, accurately reconstructs human judgment 93% of the time, whereas the next best metrics BERTScore/METEOR/ROUGE-L reconstruct accurately 80%/78%/70% of the time; A length baseline metric achieves only 60%. Notably, the reference-backed version of the newly proposed GPT-4 based metric achieves comparable (but slightly worse) performance compared to the reference-free version. Thus, we adopt the reference-free version, which additionally enables us to place the references themselves into the the Elo setup, because they are not used in the prompts.

**System-level Correlation.** We summarize the LLM's pairwise judgements using the same metrics as introduced in §4.2, Elo ratings and win rate vs. reference, but instead of using a human judge, we use our reference-free GPT-4 based metric. The results are in Table 4. Notably, among the 7 systems for which we gathered human ratings for, the automatic metric produces the same ordering compared to human evaluation ($\rho = 1.0$, $p < .01$).

**Shortcomings of proposed metric.** While the relative ranking of models produced by the automatic metric correlates strongly with the ranking produced by human judgements, the win rate vs. reference according to human judgement (Table 3) are higher overall compared to the win-rate vs. reference according to the automatic metric Table 4. One plausible explanation for this discrepancy is that GPT-4, as an evaluation model, may prefer responses that closely match its own response distribution.

**Per-category results.** In Figure 8, we plot the win-rate vs reference for the models across all the single-image instruction families. We find that there is no model that performs the best and

worst across all the instruction families. Thus, VisIT-Bench aids in highlighting the strengths and weaknesses of the instruction-following models along various real-world use-cases.

## 5   Related Work

Our work builds on prior multimodal image-text models and instruction-following benchmarks in machine learning. We provide a detailed overview of related work in §B. Multi-model modals for image-text understanding have recently emerged as powerful and useful methods for many image-language reasoning tasks [12, 15, 13, 50, 18, 14, 17, 11, 7]. Both language and multimodal models are often trained to follow language instruction, a paradigm known as "instruction following" [5, 16, 51, 14, 13, 17]. Despite the success of these approaches on existing vision-language datasets (GQA, Image Captioning [21, 52, 20]), there is no quality benchmarking dataset for multimodal instruction-following tasks that reliably replicates the way in which humans would interact with multimodal chatbots in the wild. The absence of benchmarking data impedes reliable progress assessments [53] and limits empirical evaluations of multimodal LLMs[54].

## 6   Conclusion

We present VisIT-Bench, a benchmark assessing multimodal chatbot skills. Going beyond prior efforts, VisIT-Bench's collection process centers potential real-world use cases, and 70 diverse instruction families encompassing a range of tasks from recognition to complex reasoning. Besides human-verified outputs, it features an Elo ranking aligning with human judgments. Our data reveals a performance gap between models and humans. Releasing data, code, and metrics, we aim for community engagement and believe VisIT-Bench will quantify progress and gaps in multimodal AI.

## 7   Limitations

Although VisIT-Bench covers a wide spectrum of potential use-cases, it does not incorporate every possible vision-language task. We hope to add more categories of tasks over time. In terms of dialogue, VisIT-Bench concentrates on single-turn instances with one instruction and response. This does not encompass multi-turn interactions between users and chatbots, which presents a direction for future research. Our study focuses on image-text modalities. Future extensions could expand the scope to include other modalities like audio and video, enabling a more comprehensive evaluation. Additionally, while the dataset offers a wide variety of tasks, a larger number of examples per category could provide more depth. Finally, while our GPT-4 based metric correlates well with human judgement at instance and system level, we see some evidence that the GPT-4 based metric has a stronger preference for GPT-4 based generations compared to humans. Thus, models which train, e.g., by distilling from GPT-4 outputs, may have an unfair advantage on our evaluation.

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

# Appendix

## A    License and Intended Use

The VisIT-Bench dataset, along with its various contributions such as instructions, reference outputs, and model ranking annotations, is licensed under the Creative Commons Attribution 4.0 International License (CC BY 4.0). This license applies to all the images we have directly contributed, each of which carries a public license specification in the "public images metadata" field within the dataset sheets. However, the dataset also incorporates images sourced from pre-existing collections. For these images, the original licensing terms are respected and remain applicable.

VisIT-Bench's primary purpose is to function as a dynamic benchmark that continuously evolves and evaluates instruction-following vision-language models. In the current landscape, commercial chatbots are often trained on non-disclosed and non-public datasets, which raises concerns about potential data contamination and inadvertent training on our evaluation data [8]. This risk is further highlighted by recent studies [55, 56]. To mitigate such concerns, we have chosen to withhold the complete VisIT-Bench test set from public disclosure, while still making the images and instructions available for direct download. Researchers, however, can utilize VisIT-Bench to its full potential as a dynamic benchmark by submitting their model predictions for evaluation. We will assess their models using the undisclosed test set, ensuring the ongoing evolution of the benchmark. Moreover, we are open to releasing the test data upon receiving reasonable and justified requests, particularly when additional analysis is necessary, provided that requesters agree to our non-contamination policy which prohibits the use of this data for training commercial chatbots. This approach strikes a balance between the need for robust model evaluation and the mitigation of potential data contamination.

## A    Dataset Analysis

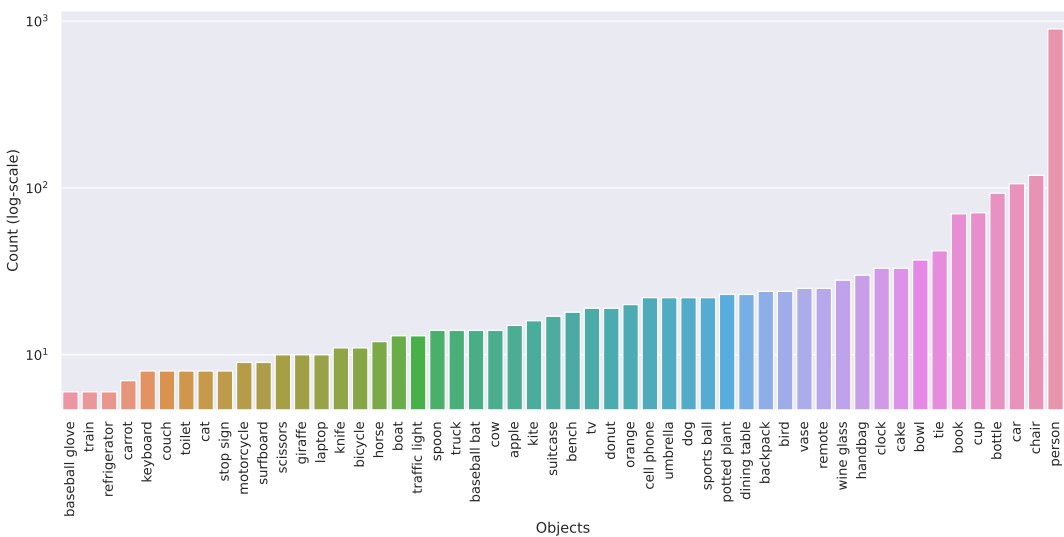

Figure 9: Count of the various COCO objects present in the VisIT-Bench images on a log-scale. The object detection was performed by Yolov5-Large [39]. We observe that the object 'person' occurs most of the time in the dataset images i.e., 875 times.

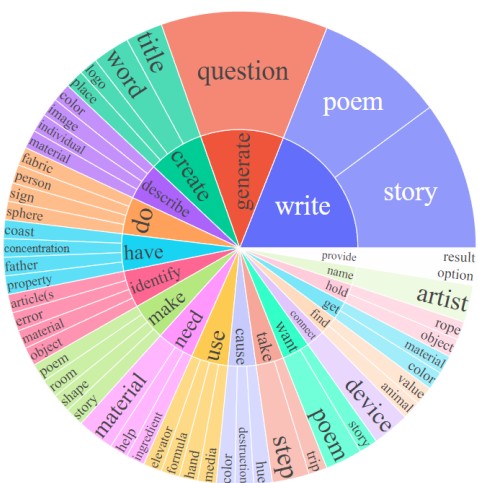

Figure 10: Most frequently occurring verbs (inner circle) and their top 4 direct nouns (outer circle) in the VisIT-Bench instructions.

## B  Related Work

**Multimodal Models for Image-Text Understanding:** Recently, the field of machine learning has experienced a rapid proliferation of new models which can perform various image-text tasks [12, 15, 13, 50, 18, 14]. This growth has been driven by several factors, including the emergence of large-scale multimodal datasets (e.g. LAION-5B [57], Multimodal C4 [11]), improved software and hardware frameworks, and advances in modality-specific models such as language models (e.g., [10]). Our work specifically evaluates models which can generate textual outputs, given one or more images, and text. Recent examples of such models include LLaVA [13], mPLUG-Owl [17], InstructBLIP, LLaMA-Adapter, Flamingo [12] and OpenFlamingo [11], PandaGPT [18], and GPT-4 [7] (which reports multimodal capabilities but has not yet seen a release of the multimodal variant).

**Instruction Following:** "Instruction-following" is an emerging paradigm for training models via language, where instead of being trained to complete only a single, fixed task (such as image classification or captioning), models are trained to follow textual instructions that describe an arbitrary task, with the aim of generalizing to novel instructions. Examples of instruction-following models include Alpaca [5], LLaMA-Adapter [16], Koala [51], InstructBLIP [14], LLaVA [13], and mPLUG-owl [17]. As the downstream capabilities of these models are influenced by the quality of the training dataset, there has also been extensive work on developing instruction-following datasets [38, 58, 59, 13, 60] .

To build powerful these models, two broad approaches have been shown to be effective. One approach focuses on leveraging existing pretrained task-specific tools such as image captioners [15], object detectors [61] and text-to-image generators [62] by either creating multimodal prompt interfaces [42, 63] or by executing LLM-generated programs [64, 65, 66]. The other approach [13, 16, 67, 45, 68, 17, 11] focuses on building a single pretrained model that can follow instructions by supervised finetuning on multimodal vision-language data.

Despite the success of both these approaches on the existing vision-language datasets e.g., VQA, GQA, Image Captioning [21, 52, 20], there is a lack of a high-quality benchmarking dataset for multimodal instruction-following tasks that reliably replicates the way in which humans would interact with multimodal chatbots in the wild. Similar to the image-text models discussed above, many instruction-following models have been released directly as open-source without undergoing peer review or thorough evaluation. As a result, the effectiveness of these models for many tasks is not well-understood.

**Benchmarks for Machine Learning:** High-quality evaluation datasets have served both to (re)assess, and to accelerate, progress on many machine learning tasks [53]. For example, our work draws particularly from the fields of computer vision and natural language processing, where benchmarking datasets have been critical drivers of progress. On the vision side, datasets such as ImageNet [69] and

CIFAR [70] have proven to be critical yardsticks of progress. On the language side, benchmarks such as SQuAD [71], SST [72], GLUE/SuperGLUE [73, 74] and more [75, 76] seen wide use. Recent work has indicated that improvements on these high-quality benchmark datasets is not the result of overfitting, and is a reliable indicator of genuine progress beyond the benchmark data [77, 78, 79, 80].

However, high-quality benchmarking datasets and evaluation methods do not yet exist for multimodal instruction-following. As a result, it is difficult to assess progress in this direction, which both reduces the field's ability to identify true breakthroughs and increases vulnerability to potential pitfalls of evaluation that have hampered progress in other areas of machine learning [53, 54].

## C    Interfaces for Collecting Human Annotations

In this section, we provide the templates we used to collect human annotations for the instruction generation (Figure 11), the dense caption generation (Figure 12), the model verification (Figure 13 and Figure 14), and the model rating (Figure 15).

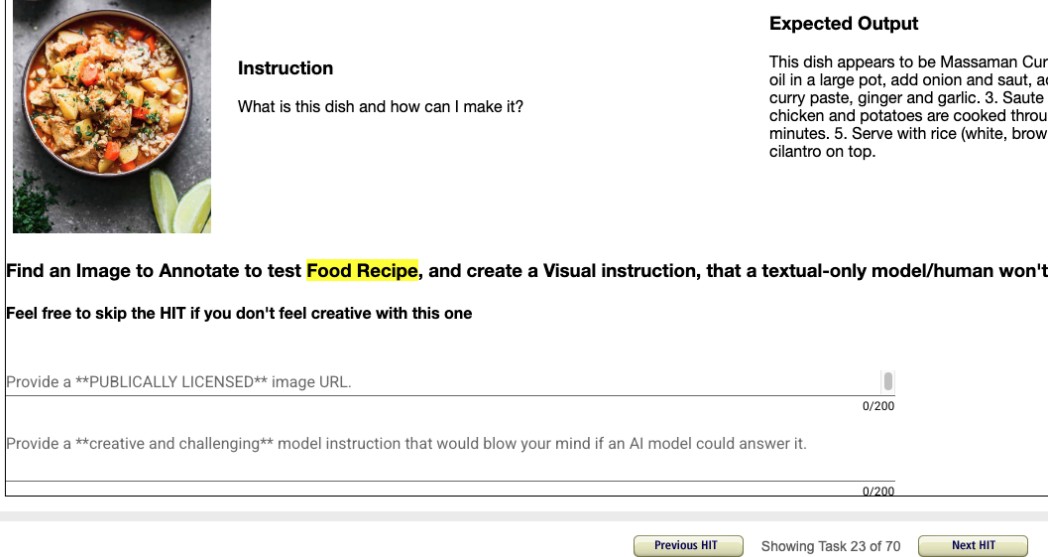

Figure 11: An interface that collects the image and corresponding instruction.

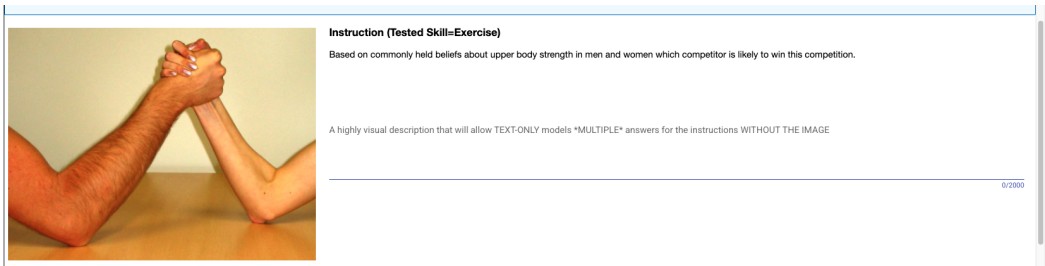

Figure 12: An interface that collects the dense caption given an image-and-instruction pair.

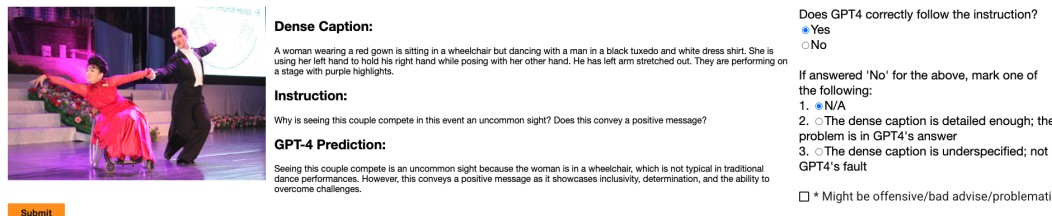

Figure 13: A sample of the model verification where the GPT4 follows the instruction correctly.

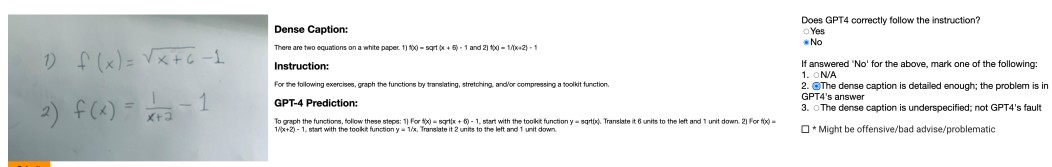

Figure 14: A sample of the model verification where the GPT4 failed to follow the instruction due to its incorrect reasoning.

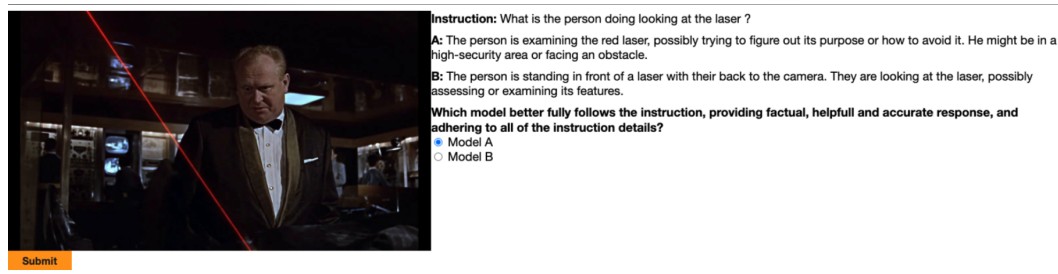

Figure 15: An interface that collects the feedback of the model rating.

# D  Existing Datasets incorporated in VisIT-Bench

In Table 5, we listed the existing datasets that are incoprated in our VisIT-Bench. Among these datasets, 15 contain a single image in each sample pair, and 10 require reasoning based on multiple images.

Table 5: List of existing datasets in VisIT-Bench, categorized as single and multiple image datasets.

| | Dataset | Topic |
|---|---|---|
| Single | VQA [21] | Visual Question Answering |
| | VCR [81] | Cognition-level Visual Understanding |
| | GD-VCR [82] | Geo-Diverse Commonsense Reasoning |
| | WHOOPS [40] | What Makes this Image Strange |
| | Newyork Caption [83] | Humor Understanding |
| | CLEVR [84] | Visual Question Answering |
| | Kilogram [85] | Tangrams Identification |
| | Harmful Memes [86] | Memes Understanding |
| | ScienceQA [87] | Science Question Answering |
| | OK-VQA [88] | Outside Knowledge Visual Question Answering |
| | AOK-VQA [89] | Outside Knowledge Visual Question |
| | AOK-VQA [89] | Question Generation |
| | VizWiz [90] | Visual Question Answering |
| | GQA [52] | Visual Question Answering on Scene Graphs |
| | TextCaps [91] | Visual Question Answering on Texts |
| Multiple | Robust Change Captioning [22] | Describing What has Change in a Scene |
| | NLVR2 [36] | Testing Visual Language Bias |
| | ImageCoDE [92] | Image Retrieval |
| | Spot-the-Diff [93] | Identifying Differences |
| | VASR [94] | Visual Analogies |
| | WinoGavil [95] | Visual Associations |
| | IRFL (Metaphor) [96] | Figurative Speech Understanding |
| | IRFL (Idioms) [96] | Figurative Speech Understanding |
| | IconQA [97] | Abstract Diagram Understanding |
| | Pick-a-Pic[98] | Text-to-Image User Preferences |

Table 6: List of skills and existing datasets in VisIT-Bench

'scienceqa', 'ocr math', 'recognition', 'okvqa', 'house plan understanding', 'nlvr2', 'gardening tips', 'textcaps', 'architectural styles', 'dressing sense', 'winoground', 'food recipe', 'paper folding', 'whoops', 'spot the diff', 'winogavil', 'imagecode', 'exercise', 'art knowledge', 'gqa', 'physical knowledge', 'contextual knowledge of events', 'home renovation', 'aokvqa', 'animals', 'vasr', 'counting', 'board games', 'solving geometry problems', 'who to call?', 'clevr', 'building materials', 'hazard identification', 'pickapick', 'astronomy', 'figurative speech explanation', 'write a story', 'gestures understanding', 'newyork', 'cultural knowledge', 'aokvqg', 'traffic sign identification', 'pop culture', 'fashion products', 'harmful memes', 'write a poem', 'vizwiz', 'guesstimate of capacity', 'location understanding', 'graph reasoning', 'vqa', 'game playing', 'differently abled', 'chemical identification', 'history knowledge', 'climate and weather understanding', 'irfl metaphor', 'human emotion recognition', 'medical', 'gd vcr', 'vcr', 'technical support', 'catchy titles', 'kilogram', 'anagrams', 'color', 'tour guide', 'directions', 'irfl idiom', 'rcc'

# E  Elo Rating

For many years, the Elo rating has been popular in ranking players in zero-sum games such as chess [25]. Recently, it has been adopted to rate large language models (LLMs) against each other on the user instructions. In this work, we adopt the same strategy to rank a set of instruction-following vision-language models, that can grow dynamically with further advances in the field.

Given two multimodal chatbots $\mathcal{C}_a$ and $\mathcal{C}_b$ with their absolute Elo rating $\mathcal{R}_a$ and $\mathcal{R}_b$, respectively. Simply put, the probability of $\mathcal{C}_a$ winning over $\mathcal{C}_b$ in a head-to-head battle is given by:

$$P(\mathcal{C}_a \text{ wins over } \mathcal{C}_b) = \frac{1}{1 + 10^{(\mathcal{R}_a - \mathcal{R}_b)/400}} \tag{1}$$

In practice, calculating the Elo rating requires us to set hyperparameters to decide the weightage for each win and loss in a head-to-head battle between two models. In our work, we use the open implementation of Elo for LLMs by FastChat at `https://github.com/lm-sys/FastChat/blob/main/fastchat/serve/monitor/elo_analysis.py`.

# F  GPT-4 Pairwise Evaluation Prompts

The specific prompts we use to extract pairwise judgements from our language model are provided in Table 16 (reference-free version) and Table 17 (reference-backed version). When applied to GPT-4 [7], these prompts usually solicit a definitive pairwise response by the model. But, in some cases, the model either produces a pairwise judgement in an unexpected format, or, refuses to issue a judgement at all. For cases like these, we issue an additional query to ChatGPT to extract an answer (or decide there is no answer) using an additional prompt, given in Table 18. If after this step there is still no definitive pairwise judgment, we call the result a tie.

A head-to-head comparison involves querying GPT-4 two times. Each query has approximately 800 input/250 output tokens (output includes chain of thought; see Supplementary for full prompt/response example). Running a head-to-head comparison between a new model and a randomly sampled existing model across the 700 instances is usually sufficient for a stable Elo estimate. In total, at current GPT-4 prices, the 700 head-to-head comparisons required to assess a new model costs 50-70 dollars.

---

`system prompt` (human authored)

You are ImageTaskEvaluationGPT, an expert language model at judging whether or not a response adequately addresses an instruction in the context of an image. More specifically, you will be given the following:

1. An image context: This will describe the contents of an image with sufficient detail to address the instruction.
2. An instruction: This is a question, an imperative request, or something similar about the image which requires a response.
3. Two responses, response A and response B: These two responses attempt to address the instruction in the context of the image. Your job is to judge whether response A or response B better. A and B are randomly ordered.

Some things to remember:

- Even though you are just a language model, the image description will be sufficiently detailed so that your judgements can be accurate.
- You are capable of judging response quality, accounting for important factors like correctness, relevance, fluency, specificity, etc.
- You think step-by-step, but ultimately respond with "Response A" or "Response B"

---

`user` (human authored)

I will describe an image to you, and provide an instruction. Then, I will give you two candidate responses that address the instruction in the context of the image: these will be labelled "Response A" and "Response B". Your job is to first reason step-by-step about which response is best in terms of accuracy, specificity, fluency, etc. After reasoning step-by-step and comparing the pros/cons of each response, in the end, respond with "Overall, Response X is better." where X is either A or B.

---

`assistant` (human authored)

Sure, please provide the image context, the instruction, and the two candidate responses, Response A and Response B. Then, I will think step-by-step and provide my ultimate judgement as to which response is better.

---

`user` (from human-authored template)

OK. Here is the image description, the instruction, and the two response options, Response A and Response B. Image context: A skier is wearing a green jacket, burgundy pants, white shoes, white gloves, and a gray beanie. There are people sitting on the background with their snowboards.

Instruction: Answer the following question based on the provided image: Are the trousers the same color as the shoes?

Response A: no

Response B: No, the trousers are not the same color as the shoes. The trousers are burgundy, while the shoes are white.

Think step-by-step and finish your response with "Overall, Response X is better." where X is either A or B.

---

`assistant` (machine generated; used to derive the prediction)

Step 1: Identify the key information from the image description.

The skier is wearing:
- Burgundy pants
- White shoes

Step 2: Assess the accuracy of each response in addressing the instruction.

Both Response A and Response B convey that the trousers and shoes are not of the same color.

Step 3: Compare specificity and detail of each response.

Response A is concise and provides a straightforward answer, but lacks detail.
Response B provides a clear, comprehensive answer that includes the color of both the trousers and shoes.

Step 4: Consider the fluency and correctness.

Both responses are fluent and grammatically correct.

Overall, Response B is better.

---

Figure 16: The prompt we use for reference-free evaluation, along with a sample instance/completion from GPT-4 [7].

Figure 17: The prompt we use for reference-backed evaluation, along with a sample instance/completion from GPT-4 [7].

---

`system prompt` (human authored)

You are FinalAnswerExtractionGPT, an expert language model at extracting multiple choice answers from written out from longer explanations. You will be given several sentences describing a thought process which should eventually contain a final answer, either A or B. Your job is to extract the final answer that the explanation arrives at.

Some things to remember:

- Keep your answers short: only output "Final Answer: X" where X is A or B
- If the explanation doesn't mention a final choice explicitly, you can output "Unknown"

...
(+ 2 `in-context examples` (human authored))
...

---

Example One: GPT-4 gives a definitive answer, but in the wrong format

`user` (GPT-4's raw pairwise judgement)

Step 1: Compare the two responses to the instruction
- The instruction asks if people are required to stop and requests an explanation.

Step 2: Assess Response A
- Response A indicates that people should stop due to the presence of a stop sign. However, it doesn't consider the fact that the sign is pressed against the bus, and the LED lights are off.

Step 3: Assess Response B
- Response B simply says "yes," acknowledging that people should stop, but it doesn't provide the explanation required by the instruction.

Step 4: Compare specificity, fluency, and other factors
- Response A is more specific and fluent but misconstrues the image description.
- Response B is accurate in that it acknowledges the stop sign but lacks an explanation.

Overall, since neither response is fully correct, it is difficult to declare a better response. However, given the fact that Response B acknowledges the prominent stop sign, I would say that Response B is slightly better, but it should also explain how the situation indicates that people should stop.
Please extract the final answer from the above text.

`assistant` (machine generated; used to derive the prediction)

---

Final Answer: Response B is slightly better, but it should also explain how the situation indicates that people should stop.

---

Example Two: GPT-4 refuses to answer.

`user` (GPT-4's raw pairwise judgement)

Step 1: Consider the image context, which states that the original thirteen colonies are shown with New Hampshire specifically highlighted in a darker green.

Step 2: Assess the accuracy of Response A, which claims that the colony shown is "Virginia." This is not accurate since the image context indicates New Hampshire being highlighted.

Step 3: Assess the accuracy of Response B, which claims that the colony shown is the Massachusetts Bay Colony. This response is also not accurate, as the image context states New Hampshire is the highlighted colony.

Step 4: In terms of accuracy, both responses are incorrect. Neither response is consistent with the image context.

Step 5: Consider other factors like fluency and specificity. Both responses seem fluent and specific in nature.

After considering these factors, none of the responses are accurate given the information provided in the image context. However, I am unable to provide an "Overall, Response X is better." judgement in this case as both responses are incorrect and neither stands out as being better than the other.

Please extract the final answer from the above text.

`assistant` (machine generated; used to derive the prediction)

---

Final Answer: Unknown.

---

Figure 18: The prompt we use to extract answers from ill-formatted GPT-4 [7] responses, along with two example outputs. In one case, GPT-4's pairwise judgement is given, but in a format different than requested. In the other case, GPT-4 "defies" the prompt, and refuses to issue a judgement because both options are bad.

