# OpenReview forum: "VisIT-Bench: A Dynamic Benchmark for Evaluating Instruction-Following Vision-and-Language Models"
_NeurIPS.cc/2023/Track/Datasets_and_Benchmarks — NeurIPS 2023 Datasets and Benchmarks Poster_

### Official Review · Reviewer_UGFt · 2023-07-04
**Unreliable evaluation using text-only GPT-4**

**Rating:** 7
**Confidence:** 4
**Correctness:** Yes
**Clarity:** Yes

**Strengths:**

1. A good research problem! It's very important to quantify the actual progress on vision-language modeling, given that most papers nowadays only provide qualitative examples.

2. A benchmark dataset has been constructed, consisting of hundreds of examples that can evaluate the instruction-following ability of vision-language models. The benchmark construction process is reasonable, including several concrete steps, like asking human annotators to provide dense captions for future evaluation.

3. Comprehensive evaluation is conducted for measuring various vision-language models.

**Additional Feedback:**

None

**Documentation:**

No, the link is invalid.

**Ethics:**

No.

**Limitations:**

This paper provides a limitation section to discuss the potential limitations of this work.

**Opportunities For Improvement:**

1. The link provided in the abstract is invalid.

2. The authors have mentioned in several places (e.g., title) the constructed benchmark is for dynamic evaluation. So why dynamic? From my understanding, a dynamic benchmark means the dataset can be dynamically changed across time to continually become more challenging to evaluate more powerful models. But all the samples in the dataset are fixed and do not possess such characteristics.

3. My major concern is about the evaluation of the responses from vision-language models, and I think this lies at the core of such benchmark evaluation work. In this work, text-only GPT-4 is used for evaluation. The results in Sec.4.4 show that the agreement between GPT-4 evaluation and Humans is only about 70%, which is quite low.

4. What is the recommended way to perform the evaluation on your benchmark? With or without the reference?

5. Could you provide an estimate of money spent to conduct a round of evaluation on your benchmark using GPT-4?

**Relation To Prior Work:**

Yes

**Summary And Contributions:**

This paper aims to quantify the progress in multimodal instruction-following models, proposing a dynamic benchmark consisting of 679 instructions. The typical characteristic is the open-ended zero-shot evaluation, which is different from existing benchmarks. The evaluation is based on GPT-4, and comprehensive evaluation is conducted across several state-of-the-art vision-language models.

---

> ### Author Response · Authors · 2023-08-18
>
> We thank the reviewer for their feedback. We are excited to see that the reviewer finds our work (a) that answers a relevant research problem to quantify the progress of instruction following models, (b) dataset construction process reasonable, and (c) evaluation comprehensive.
>
> ### Invalid Link in Abstract (Opportunities #1)
> We apologize for the confusion. The link is active now: https://visit-bench.github.io/. We will update the huggingface dataset soon.
>
> ### Clarification on 'Dynamic' Evaluation (Opportunities #2)
> Our use of the term "dynamic" pertains to the evolving nature of the Elo-leaderboard as models evolve, not the mutable size of the dataset. Recognizing the potential ambiguity, we propose altering the title to "VisIT-Bench: A Benchmark for Vision-Language Instruction Following Inspired by Real-World Use" and elucidating this distinction in the manuscript.
>
> ### Concerns on GPT-4 Evaluation (Opportunities #3)
> We thank the reviewer for bringing this up. To strengthen our arguments for the use for LLM-based auto evaluation, we perform a more extensive human agreement analysis that will be added to the revised version. We briefly explain our setup and findings here.
> For 690 pairwise instances where both candidate instances are model-generated (rather than human-verified references), we have 5 pairwise judgements from crowd-workers. For 336 pairs, there is 5/5 agreement, for 200 pairs, there is 4/5 agreement, and for 154 pairs, there is 3/5 agreement. For each metric, we measure the percent of time the metric is able to accurately reconstruct a majority vote judgment from the 5 crowdworkers. For the instances in which 5/5 annotators agree, we find that GPT4 with no reference accurately reconstructs human judgment 93% of the time, whereas the next best metrics BERTScore/METEOR/ROUGE-L reconstruct accurately 80%/78%/70% of the time.
> ### Recommended Evaluation Method – With or Without Reference? (Opportunities #4)
> In our recent human-GPT4 agreement analysis discussed above, we find that notably the reference-backed version of the newly proposed GPT-4 based metric achieves comparable (but slightly worse) performance compared to the reference-free version. Thus, we adopt the reference-free version, which additionally enables us to place the references themselves into the Elo setup, because they are not used in the prompts.
> ### Cost Estimate for Evaluation Using GPT-4 (Opportunities #5)
> A head-to-head comparison involves querying GPT-4 two times. Each query has approximately 800 input/250 output tokens (output includes chain of thought; see Supplementary for full prompt/response example). Running a head-to-head comparison between a new model and a randomly sampled existing model across the ~700 instances is usually sufficient for a stable Elo estimate. In total, at current GPT-4 prices, the 700 head-to-head comparisons required to assess a new model costs ~50-70 dollars.

---

### Official Review · Reviewer_AvKi · 2023-07-19

**Rating:** 5
**Confidence:** 3
**Correctness:** It is constructed in a sound way to me.
**Clarity:** Yes

**Strengths:**

S1: VisIT-Bench introduces an open and unique benchmark dataset for evaluating instruction-following vision-language models, providing challenging and diverse test examples that represent human intents in multimodal chatbots.

S2: Through an empirical study, publicly available model outputs are generated, accompanied by in-depth analysis of dataset diversity and assessment of model strengths and weaknesses.

S3: An automatic evaluation method utilizing human and automatic rankings based on Elo rating is employed, establishing trustworthiness through instance-level agreement analysis with human preferences.

**Additional Feedback:**

[11] and [23] refer to the same reference.

**Documentation:**

Yes

**Ethics:**

No ethical concerns I found.



**Limitations:**

see the above



**Opportunities For Improvement:**

O1: Some crucial references, such as [1] and its related surveys/benchmarks, have not been discussed in the current work.

[1] Yin et al. A Survey on Multimodal Large Language Models

O2: Although the paper states the existing works lack a formal evaluation, many multimodal capabilities cater to users' applications, the qualitative evaluation (e.g., based on user study) may not be an issue.

O3: Inadequate elaboration on the roles of different steps in Fig. 2's caption and the main text, lack of detailed information in the figure, and unclear meanings of certain operations, along with a data collection strategy similar to existing works [11][23].





**Relation To Prior Work:**

Some prior works are not discussed, see O1.

**Summary And Contributions:**

Recent advancements in instruction-following vision-language models have led to the rise of large-scale multimodal chatbots, but the lack of comprehensive evaluation in understanding human instructions and delivering useful outputs, is addressed by the introduction of VisIT-Bench, a robust benchmark with diverse real-life vision-language tasks, enabling in-depth assessment of conversational abilities and incorporating an Elo-based ranking system aligned with human evaluations, demonstrating reliability and providing a publicly available benchmark for integrating and evaluating new models.

---

> ### Author Response · Authors · 2023-08-18
>
> We thank the reviewer for their diligent efforts. We are motivated to see that the reviewer finds our work (a) unique and challenging for new instruction following VL models, (b) in-depth in model evaluation, and (c) trustworthy.
>
> ### Inclusion of the Work by Yin et al. and Related Benchmarks (Opportunities #1)
> We've incorporated Table 1 in the revised manuscript, contrasting VisIT-Bench with related benchmarks. Notably, the Yin et al. paper was post-NeurIPS, which explains its initial omission. It has now been duly incorporated
>
> ### Necessity of a Formal Evaluation (Opportunities #2)
> Our benchmark provides standardized and objective criteria to evaluate the performance of instruction following VL models.
> Such benchmarks help ensure fair and unbiased comparisons between different AI models and techniques.
> When a model performs poorly on a specific instruction family, it indicates potential weaknesses or limitations in the approach. This insight allows researchers to refine their methods and uncover areas for improvement, driving progress in the field.
> Since the track is Datasets and Benchmarks, we believe that formal evaluation via benchmark datasets is in scope for that.
>
> ### Clarifications on Approach vs. Existing Works [11][23] (Opportunities #3)
> Our approach, while seemingly analogous, diverges significantly from Visual Instruction Tuning (VIT) cited in [11,23]. Whereas VIT offers a cursory evaluation using GPT-4 on a limited set of COCO images, VisIT-Bench encompasses 678 instances across 70 instruction-families, rooted in authentic human instructions.
> Unlike VIT, our evaluation metrics are more comprehensive, encompassing human evaluations, GPT-4 pairwise comparisons with high human agreement, win-rates, and Elo ratings of various instruction-following models.
> It's also important to note that our dataset includes multi-image instances, a feature not present in VIT.
>
> The clarity gaps in Figure 2 have been addressed in the revision. We have enriched the caption, providing a more granular elucidation of each step, making it more intuitive for readers.
>
> ### Duplicate Reference Issue [11] and [23] (Additional Feedback #1)
> We thank you for highlighting the [11] and [23] oversight. This has been rectified in the revision.

---

> > ### Author Response · Authors · 2023-08-26
> >
> > Dear Reviewer AvKi,
> >
> > Thank you once again for your comprehensive review and insights on our VisIT-Bench paper.
> >
> > On August 18th, we posted our response and a revised manuscript. We've made several enhancements to our manuscript. Specifically in response to your feedback, we added Table 1 for benchmark comparisons, clarified our approach versus existing works cited as [11][23], and corrected the duplicate reference issue.
> >
> > We would be very grateful if you get a chance to respond and/or adjust your score. We would be happy to provide any additional clarifications if requested.
> >
> > As the rebuttal period concludes this Tuesday, your final thoughts would be immensely valuable to us.
> >
> > Best regards,
> > The VisIT-Bench Team

---

### Official Review · Reviewer_ywz9 · 2023-07-20
**Review on VisIT-Bench: A Dynamic Benchmark for Evaluating Instruction-Following Vision-and-Language Models**

**Rating:** 6
**Confidence:** 4
**Correctness:** It is somewhat sound.
**Clarity:** The paper is well written and easy to…

**Strengths:**

1. It provides both automatic and Elo-based human evaluation for evaluating the instruction-following ability of multimodal large language models. The benchmark covers diverse real-life vision-language instructions across 70 tasks.
2. The way to add detailed dense caption and verified reference outputs using language-only GPT4 model for evaluation is good and proves to be more effective.

**Additional Feedback:**

1. Can you provide the final data statistics of the benchmark dataset? how about the actual size of the benchmark dataset?
2. It is better to see how different models perform in different tasks or categories, except for only a global comparison result.

**Documentation:**

Somewhat sufficient.

**Opportunities For Improvement:**

1. The details of how to annotate dense caption and the labeling rules are not provided. Can you provide the details? How many examples are annotated with dense captions?
2. The dense captions may not contain the information of relative spatial relation and the relation of the detected objects. I am wondering whether the evaluation has considered the evaluation of this type of question. Besides, the MLLM may have serious hallucination problem, I am wondering if the language-only GPT4 can detect this type of problem.

**Relation To Prior Work:**

Yes

**Summary And Contributions:**

This paper introduces a robust evaluation benchmark for evaluating the instruction-following ability of multimodal large language models. It incorporates the dense caption and verified reference outputs for automatic evaluation of  multimodal large language models via text-based GPT4. Besides, it also provides an Elo-based ranking system to establish a leaderboard for multimodal chatbots. The results show that Elo-rankings approaches show strong agreement with human evaluations.

---

> ### Author Response · Authors · 2023-08-18
>
> We thank the reviewer for their feedback. We are pleased to learn that they appreciate our human and automatic ELO-based evaluations. Additionally, we're glad that our approach to adding detailed captions and referencing outputs has proven effective.
> ### Annotating Dense Captions (Opportunities #1)
> Details of dense caption annotation and labeling rules can be found in Figure 2 and Section 2.1. Additionally, Appendix B provides a comprehensive view of our mechanical turk layouts. For clarity, all 678 instances in VisIT-Bench are accompanied by dense captions.
> ### Addressing Spatial Relations and Hallucinations Issues (Opportunities #2)
> We concur on the importance of understanding spatial relationships in tasks. VisIT-Bench incorporates such tasks sourced from datasets like CLEVR, NLVR2, and select VQA instances. When the task necessitated relational comprehension, this was meticulously incorporated into the dense caption. The high accuracy (91.5%) demonstrated by text-only models (Table 2) underlines the effectiveness of our dense captions.
> Concerning MLLMs' potential hallucination issues, if an instance of this arose, it would affect the text-only GPT-4's performance, yet our results suggest our dense captions manage to guide the model effectively.
> ### Benchmark Dataset Statistics (Additional Feedback #1)
> VisIT-Bench comprises 678 entries, each containing an image, instruction, caption based on the instruction, a GPT-4 response suggestion, and a label verifying the accuracy of the response. Out of these, 592 instances received human verification, ensuring the GPT-4 response's accuracy. We also publish the additional 5,000 human preference data for model responses [here](https://github.com/mlfoundations/VisIT-Bench/tree/main#human-preference-annotations).
> ### Task-Specific Model Performance (Additional Feedback #2)
> We agree with the reviewer's point on task-specific model performances. Figure 8 is dedicated to highlighting how different models fare across various instruction families. Interestingly, no model emerges as a consistent top-performer across all categories, suggesting significant room for advancement in current models.

---

> ### Author Response · Authors · 2023-08-26
>
> Dear Reviewer ywz9,
>
> Thank you for your thoughtful and comprehensive review of our VisIT-Bench paper. Your feedback has been highly constructive.
>
> On August 18th, we submitted a revised manuscript along with our detailed responses to your concerns. To quickly recap, we have elaborated on the dense caption annotation process and the labeling rules in Figure 2 and Section 2.1. We also addressed your questions about spatial relations and the potential hallucination issues with MLLMs. Finally, we included more data statistics and task-specific performance details in line with your suggestions.
>
> We would be very grateful if you get a chance to respond and/or adjust your score. Your final thoughts would be particularly valuable as the rebuttal period closes this Tuesday.
>
> Best regards,
> The VisIT-Bench Team

---

### Official Review · Reviewer_Fmx9 · 2023-07-21

**Rating:** 7
**Confidence:** 4
**Clarity:** The paper is well written and easy to…

**Strengths:**

1. The paper incorporates an Elo-based ranking system to establish a leaderboard for multimodal chatbots. This system, utilizing GPT-4 to provide pairwise preferences based on instructions and candidate outputs, enables the evaluation of multimodal models in an automated manner.
2. The instances in VisIT-Bench are categorized into specific "skills," which better captures the ability of models to follow instructions of various types.

**Additional Feedback:**

NA

**Correctness:**

The proposed dataset is constructed using both LLM-based and human-based methods, ensuring a robust approach. However, there are some aspects that remain unclear:

- In Line 39, the selection process for the 10 images is not explained.
- In Line 44, it is unclear how the regions in VCR are processed.

**Documentation:**

Yes.

**Ethics:**

No.

**Limitations:**

See above.

**Opportunities For Improvement:**

1. The data size of VisIT-Bench, approximately 700 instances, is relatively small compared to other visual LLM benchmarks (e.g., M3Exam [1] with over 12k instances, M3IT [2] with 2.4m instances).
2. It is unclear whether the inclusion of "dynamic" datasets in the benchmark is beneficial. The authors mention integrating more datasets dynamically for comprehensive evaluation in the future (Line 320), but this may introduce evaluation instability. The term "dynamic" suggests that the authors may not be fully prepared for the release of this benchmark.

[1] M3Exam: A Multilingual, Multimodal, Multilevel Benchmark for Examining Large Language Models. Zhang et al.

[2] M³IT: Multi-Modal Multilingual Instruction Tuning Dataset. Li et al.

**Relation To Prior Work:**

The paper clearly discussed how this work differs from previous contributions.

**Summary And Contributions:**

This paper presents VisIT-Bench, a benchmark for evaluating instruction-following visual large language models. VisIT-Bench consists of ~700 VL instructions, spans 70 instruction families, encompassing a range of tasks from recognition to complex reasoning. This benchmark offers human-verified reference outputs for all test examples and an Elo-based ranking system for multimodal chatbots. The authors also evaluate prevelant visual LLMs like InstructBLIP, MiniGPT-4 based on VisIT-Bench, and show that their performance still trail behind text-only GPT-4 and human performance.

---

> ### Author Response · Authors · 2023-08-18
>
> We appreciate the reviewer's insightful feedback and acknowledgment of the innovation in our Elo-based ranking system, as well as the categorization of instances in VisIT-Bench into distinct "skills." Such positive comments validate our objectives with the benchmark.
>
> ### Data Size of VisIT-Bench Compared to Other Benchmarks (Opportunities #1)
>
> Thank you for highlighting the data size concern. We've now incorporated a direct comparison in Table 1 of our revised manuscript, setting VisIT-Bench against other benchmarks.
>
> Though smaller than M3Exam and M3IT, VisIT-Bench focuses on depth over quantity. Our inclusion of 70 diverse instruction families ensures broad coverage of real-world challenges. Notably, both M3Exam and M3IT were released post our NeurIPS submission, yet when compared:
>
> VisIT-Bench evaluates 10 models vs. M3IT's 4.
> We feature 70 skill sets, surpassing M3IT's 13.
> Our evaluation framework leverages both human insights and GPT-4's automated prowess, diverging from M3IT's focus on dataset-specific assessment.
> We provide win-rates and Elo ratings, absent in M3IT, for a comprehensive performance view.
>
> Our development process involved qualified annotators and multiple steps: instruction generation, instruction-condition caption generation, and model output evaluation.
>
> Despite its compactness, models find VisIT-Bench challenging. Our GPT-4 auto-evaluation aligns well with human judgments and outperforms established metrics like BERTScore, METEOR, and ROUGE-L. Additional details and visuals will be included in our revision.
>
> ### The "Dynamic" Nature of VisIT-Bench (Opportunities #2)
>
> We recognize the potential ambiguity the term "dynamic" may introduce. Our intention was to highlight the adaptability of the Elo-leaderboard, not a fluctuation in dataset content or size. To dispel any confusion, we've decided to retitle our work to "VisIT-Bench: A Benchmark for Vision-Language Instruction Following Inspired by Real-World Use," ensuring clarity in the manuscript.
>
> ### Image Region Processing in VCR (Correctness #2)
> For clarity, we composed instruction-conditioned dense captions to ensure they contain adequate detail to adhere to the given instruction. If an image region was pivotal to comprehending the instruction, we incorporated pertinent information into the caption.

---

> > ### Comment · Reviewer_Fmx9 · 2023-08-21
> > **Response to the rebuttal**
> >
> > Thanks for the clarification. The comparison between VisIT-Bench and other benchmarks enhances the presentation quality of the work, accordingly, I will increase my score to 7.

---

### Author Response · Authors · 2023-08-18
**General Response to Reviewers**

We appreciate the reviewers for recognizing our implementation of an Elo-based ranking system for the evaluation of multimodal models (Fmx9), the breadth and diversity of real-life vision-language instructions in VisIT-Bench spanning 70 tasks (ywz9), and VisIT-Bench's distinct role in gauging the instruction-following ability of vision-language models mirroring human intents (AvKi). The innovative approach generating detailed dense captions and using GPT4 to follow the instructions received notable acknowledgment (ywz9, UGFt).

Below, we address each reviewer's comments and questions in detail.

---

> ### Author Response · Authors · 2023-08-20
>
> We have revised the paper based on reviewers' invaluable feedback, highlighting changes in red and prefixing them with the reviewer ID (\textcolor{red}{[reviewer-id]}).

---

### Decision · Program_Chairs · 2023-09-22

**Decision:**

Accept (Poster)

**Comment:**

This paper was reviewed by four experts with mixed scores. AC feels that this work makes valuable and timely contributions to benchmark the performances of VLM models. The reviewers did raise some valuable concerns that should be addressed in the final camera-ready version of the paper.